# Non-Suicidal Self-Injury: An Observational Study in a Sample of Adolescents and Young Adults

**DOI:** 10.3390/brainsci11080974

**Published:** 2021-07-23

**Authors:** Emilia Matera, Mariella Margari, Maria Serra, Maria Giuseppina Petruzzelli, Alessandra Gabellone, Francesco Maria Piarulli, Assunta Pugliese, Anna Rita Tassiello, Federica Croce, Caterina Renna, Anna Margari

**Affiliations:** 1Department of Biomedical Sciences and Human Oncology, University Hospital “A. Moro”, Piazza Giulio Cesare 11, 70100 Bari, Italy; alessandragabellonee@gmail.com; 2Department of Basic Medical Sciences, Neuroscience and Sensory Organs, University Hospital “A. Moro”, Piazza Giulio Cesare 11, 70100 Bari, Italy; mariella.margari@hotmail.it (M.M.); maria.serra.13@gmail.com (M.S.); maria.petruzzelli@uniba.it (M.G.P.); piarullif@gmail.com (F.M.P.); silegra@hotmail.it (A.P.); annarita.tassiello@libero.it (A.R.T.); fedix90@live.it (F.C.); margarianna2@gmail.com (A.M.); 3Center for Treatment and Research on Eating Disorders Mental Health Department AL, Via Miglietta 5, 73100 Lecce, Italy; caterinarenna@gmail.com

**Keywords:** non-suicidal self-injury, adolescents, young, emotion regulation, sensation seeking, Craving, impulsivity

## Abstract

Non-Suicidal Self-Injury (NSSI) is the self-inflicted destruction of body tissues without suicidal intent with a prevalence of 1.5% to 6.7% in the youth population. At present, it is not clear which emotional and behavioral components are specifically associated with it. Therefore, we studied NSSI in a clinical sample of youth using the Ottawa Self-injury Inventory and the Barratt Impulsiveness Scale 11. The Mann–Whitney test was used to compare the numerical responses provided to the tests. We found 54 patients with NSSI, with a mean age of 17 years. Scores were analyzed in the total sample and in four subgroups. In the total sample, Internal Emotion and External Emotion Regulation, Craving, Non-Planning and Total Impulsivity were significantly associated with NSSI. There were statistically significant differences in Craving between patients with multiple NSSI episodes, suicide attempts and multiple injury modes and patients of other corresponding subgroups, in Internal Emotion Regulation, Sensation Seeking and Motor Impulsivity between NSSI patients with suicide attempts and no suicide attempts, and in Cognitive Impulsivity between NSSI patients with multiple injury modes and one injury mode. It is necessary to carefully evaluate the components underlying NSSI in order to activate personalized treatment options.

## 1. Introduction

Non-Suicidal Self-Injury (NSSI) is the voluntary and self-inflicted destruction of body tissues in the absence of suicidal intents with no socially sanctioned purposes. The main feature of NSSI is recurrent self-infliction of superficial injuries towards the own body, commonly associated with an immediate sense of relief. The wounds are often inflicted with knives, needles, razors or other sharp objects. Commonly injured areas are the thighs’ front and the forearms’ dorsal side. Methods used are cutting, stabbing, superficial burning with a lit cigarette butt or burning the skin by repeatedly rubbing the skin with an eraser [1].

NSSI is frequently observed between the ages of 12 and 14 years and ranges from 1.5% to 6.7% in community samples of children and adolescents [1]. NSSI is also a significant concern for children and adolescents’ clinicians because it has a significant risk of relapse and a strong association with suicidality and psychiatric disorders [2,3,4,5].

In recent years, a strong debate whether NSSI could be just a symptom in the context of a broader psychopathological picture or a specific diagnostic category has been conducted. In fact, NSSI has been included in the third section of the Diagnostic and Statistical Manual of Mental Disorders—Fifth Edition (DSM 5) which groups the conditions requiring further investigations [6].

Several authors have reviewed the empirical research about self-injury, examining its underlying causes and clinical phenomenology as well as the effects of this behavior on the physiological affective arousal. To date, it is widely thought that there is an emotion-regulation function of NSSI, and several research results support the existence of an emotion dysregulation trait among people who engage in this behavior [7,8,9,10,11,12]. Emotional regulation is a multifactorial construct that includes the awareness, understanding and acceptance of emotions, the ability to inhibit impulsive behavior related to emotional distress and the willingness to avoid activities that may trigger negative emotions such as tension, depression or anger. Conversely, emotion dysregulation is defined by a different combination of decreased emotional awareness, inadequate emotional reactivity, intense experience and expression of emotions, emotional rigidity and cognitive reappraisal difficulty [5]. These key components lead to maladaptive emotion regulation strategies with manifold implications for clinical practice. Empirical evidence suggests that NSSI is commonly performed as an emotion regulation strategy, as it often decreases the perception of negative affect in many different ways.

A theoretical model proposed by Nock (2009) hypothesized that both general and specific factors might increase the risk of NSSI. More distal risk factors, such as childhood abuse and genetic predispositions to high emotion reactivity, may be considered no-specific vulnerability factors predisposing a subject to respond to stressful life events in a maladaptive manner. Several more specific factors could explain why some people specifically use NSSI to regulate one’s emotional/cognitive negative experiences and to communicate with others (e.g., observation of social patterns, self-punishment, communication of high intensity/high-cost behaviors, pain analgesia, implicit identification in self-injurious behavior) [13].

Moreover, according to Cyders and Smith’s Theory of Urgency (2008), some subjects are more prone to impulsive behaviors when experiencing extreme negative affect, increasing the risk of potentially harmful behaviors [14]. Individuals who are highly impulsive and with negative urgency may be at high risk for NSSI engagement since they may be highly motivated to obtain the immediate short-term gains of NSSI in the context of negative emotions, with less concern for the long-term consequences of NSSI engagement. Impulsivity could thus increase vulnerability to engage in readily available but maladaptive behaviors, such as self-injury, to moderate negative affective states and the perceived efficacy of this strategy could lead to negative reinforcement of self-injury. Furthermore, Andover and Morris (2014) emphasized that NSSI not only helps to reduce negative emotions but also tends to generate positive emotions, which reinforces the use of such behavior [15]. This is consistent with the hypothesis of the addictive feature of NSSI, considering that this maladaptive behavior is characterized by dependence and loss of control, with a high risk of recurrence or engaging in similar behaviors despite the negative consequences [4].

Although several studies provided evidence of addictive features of NNSI, many youths reported self-harm behavior anecdotally and some differences have been identified between Craving in NSSI and drug use, so the question of whether NSSI can become an addictive behavior remains an unresolved conceptual debate.

On the other hand, a better understanding of the potential factors playing a role in perpetuating NSSI has significant clinical implications, considering that the maintenance over time of NSSI behavior is a relevant risk factor for suicidal behavior in adolescence [16].

We hypothesized that a different combination of dimensions related to emotion dysregulation, impulsivity and addictive behavior might be associated with a different level of severity of NSSI, with specific regard to the persistence over time and to the risk of suicidality. To test this hypothesis, we selected a sample of adolescents and young adult psychiatric inpatients with a history of NSSI, and we searched for motivations to engage in self-harm behavior, dimensional traits of impulsivity and addictive features of NSSI.

The principal aims of this paper were:1.To collect information on general clinical features of NSSI behavior (duration, frequency, methods, etc.) using the Ottawa Self-Injury Inventory (OSI), a comprehensive self-report measure of non-suicidal self-injury. Then, we assessed the four-factor structure of the OSI functions items, consisting of four categories of motivations underlying NSSI behavior (Internal Emotion Regulation (IER), Social Influence (SI), External Emotion Regulation (EER), Sensation Seeking (SS)), as well as the single-factor structure of Addictive features items (Craving (C));2.To evaluate the unidimensional latent trait of impulsivity, measuring the Barratt Impulsiveness Scale-11 (BIS 11) [17] total score and the multidimensional construct of impulsivity corresponding to the three BIS-11 subdomains (Cognitive, Motor and Non-Planning Impulsivity (CI, MI, NPI));3.To study the potential association between the OSI functions and Addictive features items, the BIS-11 dimensions (total score and subdomains, respectively) and indicators of NSSI severity, including frequency, number of injury sites, method versatility as well as suicidal behaviors.

## 2. Materials and Methods

We searched for the presence of NSSI in a sample of adolescents and young adult inpatients hospitalized for acute psychopathological disorders in the Operative Units of Child Neuropsychiatry and Psychiatry of the University Hospital of Bari, Italy, between January 2018 and September 2020.

Patients with an intellectual disability or other conditions that would not allow active participation in the assessment procedures were excluded from this study.

Neuropsychiatric diagnoses were made by a child neuropsychiatrist/psychiatrist according to DSM 5 criteria, with the help of medical history collection and clinical observation. Socio-demographic and clinical data were recorded in a database.

All participants underwent the administration of the following standardized psychometric protocols:

1. OSI [18]: it is a self-administered questionnaire consisting of 27 items about cognitive, affective, behavioral and environmental aspects of non-suicidal self-injury. The scales examine the following elements: frequency of NSSI thoughts and episodes, reasons for initiating and continuing self-injury functions of NSSI behavior, addictive characteristics as the level of proneness in stopping the behavior and other features related to NSSI nature. There are also items related to the usefulness of eventual previous treatments.

The test does not provide a total score or a cut-off, but the domain with the highest mean score indicates the primary motivation for initiating and/or continuing the activity.

The OSI has been shown to be valid and reliable with internal consistency values ranging from 0.67 to 0.87 in a university sample of young adults and is suitable for use with clinical samples of adolescents. An adapted OSI Italian version from Nixon et al. [19] was used.

Due to the structure of the questionnaire (presence of items that include both quantitative and qualitative responses) and in accordance with the aims of our study, we considered only the following OSI items:n. 2 (“How often in the past 6 months have you actually injured yourself, without the intention to kill yourself?”) with the score range between 0 (not at all) to 4 (every day);n. 4 (“Have you ever made an actual attempt to take your life?”) with “yes” or “no” as options;n. 12 (“What areas of your body did/do you injure”) in which the most common places of injury can be listed;n. 13 (“How did/do you injure yourself?”), in which it is possible to list the most common types of injuries;n. 14 (“Why do you think you started and if you continue, why do you still self-injure (without meaning to kill yourself?”)), with scores ranging from 0 (never a reason) to 4 (always a reason) and expressed as IER (scoring between 0 and 24, that is the sum of the subitems 4, 6, 9, 14, 16, 18), SI (scoring between 0 and 28, that is the sum of the subitems 3, 9, 10, 11, 13, 15, 21), EER (scoring between 0 and 12, that is the sum of the subitems 1, 12, 20), SS (scoring between 0 and 16, that is the sum of subitems 2, 7, 22, 23);n. 20 (“Since you started hurting yourself, you realized that”), where the sub-scores range from 0 (never) to 4 (always) and the total score, variable between 0 and 28, indicates C;

2. BIS 11 [17]: it is a self-administered tool that assesses impulsivity as a behavioral or personality variable. The structure of BIS allows to identify six first-order factors (attention (items 5, 9 *, 11, 20 *, 28), motor behavior (items 2, 3, 4, 17, 19, 22, 25), self-control (items 1 *, 7 *, 8 *, 12 *, 13 *, 14), cognitive complexity (items 10 *, 15 *, 18, 27, 29 *), perseverance (items 16, 21, 23, 30 *), cognitive instability (items 6, 24, 26)) and three second-order factors: CI (obtained with the sum of attention and cognitive instability), MI (obtained with the sum of motor behavior and perseverance) and NPI (obtained with the sum of self-control and cognitive complexity). The current version of BIS-11 consists of 30 items whose scores are assigned on a 4-point scale ranging from 1 (rarely/never) to 4 (almost always/always). Items with an asterisk (*) are scored inversely, from 4 (never/rarely) to 1 (almost always/always). The total score (TI) ranges from 30 to 120 and provides a quantitative assessment of impulsivity, which is the sum of CI (minimum score: 8, maximum score: 32), MI (minimum score: 11, maximum score: 44) and NPI (minimum score: 11, maximum score: 44). Patton et al. reported internal consistency coefficients for the BIS 11 total score ranging from 0.79 to 0.83 for separate populations of under-graduated students, substance-abuse patients, general psychiatric patients and prison inmates [17]. The Italian validated version of BIS was used [20].

The OSI and BIS 11 scores were analyzed both in the total sample and in the following four subgroups:1.NSSI patients with an occasional episode vs. NSSI patients with multiple episodes;2.NSSI patients with no suicide attempts vs. NSSI patients with suicide attempts;3.NSSI patients with a single site of injury vs. NSSI patients with multiple sites of injury;4.NSSI patients with single injury mode vs. NSSI patients with multiple injury modes.

## 3. Statistical Analysis

All variables were subjected to descriptive analysis for the socio-demographic and clinical data. Means and standard deviations (SD) were calculated for the following variables: age and numerical responses provided to the BIS 11 and OSI’s items 14 and 20.

The non-parametric Mann–Whitney test was used to compare the mean values of IER, SI, EER, SS, C, CI, MI, NPI, TI between the four subgroups.

The non-parametric Mann–Whitney test was also used to check for age, the age of onset and duration of illness-related confounding factors and homogeneity between the same groups. Data were processed using STATA 11 software (StataCorp LP, College Station, TX, USA) for Mac OS. The significance level was set at *p*-value < 0.05.

## 4. Results

Socio-demographic and clinical features of the sample are shown in Table 1.

In a sample of 3944 patients hospitalized for acute neuropsychiatric disorders, we found 54 patients with NSSI, 44 females and 10 males, with a mean age of 17.07 years (SD ± 4.09). The mean age at NSSI onset, determined from OSI, was 13.17 years (SD ± 2.5); female patients reported an earlier onset of self-injurious behavior (mean age 13.18 years, SD ± 2.2) than males (mean age 14.8, SD ± 3.3). In addition, 37% of patients presented Depressive Disorders (Major Depressive Disorder, Disruptive Mood Dysregulation Disorder), 14.8% presented Personality Disorders (Bordeline Personality Disorder) and 22.2% of the sample presented Trauma and Stressor Related Disorders. One or more psychopathological comorbidities were present in 61% of the sample.

### Ottawa Self-Injury Inventory and Barratt Impulsiveness Scale 11

The mean values and SD of the OSI and BIS 11 items are shown in Table 2 and Table 3, respectively.

Items 14 and 20 of the OSI showed higher mean scores for IER, EER and C subtests. In the BIS 11, there were higher mean scores for the items NPI and TI.

Patients with multiple NSSI episodes showed higher C mean scores with a statistically significant difference (*p* = 0.001) compared to NSSI patients with one occasional episode (*n* = 36).

NSSI patients with suicide attempts had higher IER (*p* = 0.001), SS (*p* = 0.023), C (*p* = 0.006) and MI (*p* = 0.025) mean scores with a statistically significant difference compared to NSSI patients without suicide attempts.

NSSI patients with multiple injury modes showed higher C (*p* = 0.029) and CI (*p* = 0.049) mean scores with a statistically significant difference compared to NSSI patients with a single injury mode.

Furthermore, no statistically significant differences were found between NSSI patients with a single injury site and NSSI patients with multiple injury sites. However, higher mean C scores were found in the subgroup with multiple injury sites (see Table 4, Table 5, Table 6 and Table 7).

## 5. Discussion

In this study, we analyzed NSSI in a sample of adolescents and young adults with various neuropsychiatric disorders with a focus on the main emotional, behavioral and addiction-related components of NSSI and their relationship to different clinical NSSI phenotypes sorted by the number of NSSI episodes, number of injury sites, number of injury modes and association with suicide attempts.

### 5.1. Socio-Demographic and Clinical Features of the Sample

The NSSI patients in our sample represented 1.36% of patients hospitalized for acute psychopathological disorders from January 2018 to September 2020. Otherwise, the incidence found in the general population is around 2–5% [21,22]. This difference could be consequent to different reasons.

The first reason is related to the nature of our sample, which consisted of hospitalized patients, so that we could hypothesize an under-representation of the phenomenon. In addition, young people who act self-injuring behaviors are unlikely to seek clinical help probably because of the stigma or, in reverse, for the positive meaning that the person involved gives to NSSI [6].

In our sample, there was a higher prevalence of NSSI in female patients than in male ones. These data are consistent with most of the available literature [23,24,25], although some articles reported an inversion of NSSI ratio between males and females when considering the type of activation/repetition of the behavior, the severity and type of the associated psychopathology and certain cultural contexts [26,27,28,29]. In our study, the mean age of the enrolled patients was 17.07 years (SD ± 4.09), with a mean age at NSSI onset of approximately 13 years and an earlier NSSI onset in females. After the age of 18 years, the frequency of self-injurious behavior decreased. Furthermore, the association between NSSI, adolescence and female gender is consistent with the importance of the role that biological components play. The rapid maturation of the limbic system and the increased sensitivity of the serotonergic system and glucocorticoids, which in turn are influenced by a specific hormonal environment, could represent the specific biological pathways involved in this gender-related difference [30,31,32]. In fact, sex hormones and pubertal timing could be involved in the pathophysiology of these disorders through the modulation of the neuroendocrine system. Moreover, sex-specific hormonal differences in estrogen-testosterone ratio, variability in the timing of physical development and changes associated with menarche and menstruation could also represent possible biological mechanisms involved, explaining the gender difference found in NSSI behavior [33,34,35,36]. Structural and functional brain alterations could also represent a possible neurobiological key vulnerability in developing maladaptive coping strategies in adolescents. Compared to healthy controls, adolescents and adults with NSSI showed diffuse abnormalities of amygdala circuits (frontal lobe, supplementary motor area, dorsal anterior cingulate and occipital lobe) and deficit in white matter microstructure in the uncinate fasciculus, cingulum, bilateral superior and inferior longitudinal fasciculi, anterior thalamic radiation, callosal body and corticospinal tract. On the one hand, such alterations seemed to worsen with frequent recurrence of NSSI behaviors and, on the other hand, could have a role as neurobiological targets for the individuation and management of specific therapeutic interventions [37,38,39].

Our sample showed that NSSI was most common in patients with depressive disorders, personality disorders and trauma and related pathologies. Moreover, 61% of our patients had comorbidity with one or more psychopathologies, suggesting that many clinical conditions may be associated with NSSI, as reported in the literature [40,41,42]. In the end, it is important to underline the epidemiological data that our sample was predominantly composed of female and adolescent patients. Mental disorders, together with accidents and reproductive and sexual sphere diseases, are the main causes of morbidity and mortality in the age group between 10 and 19 years [43]. Moreover, due to the gender-related susceptibility of many neuropsychiatric disorders, depressive disorders predominate in women [44].

### 5.2. NSSI Emotional and Behavioral Components

In our sample, the most involved dimensions in perpetuating NSSI were IER, EER and C. High NPI and TI scores were also found. There is evidence in the literature that self-injuring young people show significantly higher levels of non-acceptance/regulation/understanding of emotions, both intrapersonal (escaping negative emotions, replacing mental or emotional pain with physical pain, seeking emotions to combat the feeling of anhedonia or affective flattening) and especially interpersonal (communicating malaise, seeking help and escaping difficult situations) [22,45,46]. In addition, an association seems to exist between NSSI and traumatic/stressful events such as experiences of abuse and interpersonal difficulties with peers and family [47,48]. In fact, this finding is also evident in our sample, in which approximately 22% of patients received a primary diagnosis of Trauma and Stressor Related Disorder.

The study of the association between NSSI and impulsivity is complex due to numerous elements such as the type of conceptualization and the method of evaluation; therefore, the results are not always consistent between different authors [49,50,51,52,53,54]. Studies using BIS-11 showed a total score associated with NSSI only in males, whereas the Chinese version BIS factor 3 (seeking novelty and action without thinking) was associated with non-suicidal self-injury in both males and females [55]. Emotional contexts involving poor acceptance/negative criticism, real or imagined, in social relationships [56] would favor the link between NSSI and impulsivity [53,54,55,56,57,58,59,60,61]. This relationship also seems to be explained by anatomical correlates such as reduced functional connectivity between the left orbitofrontal cortex and the right parahippocampal gyrus [41], an increase in the activation of the cingulate cortex and a reduction in the activation of the dorsolateral prefrontal cortex [62].

About the relationship between NSSI and C, the available literature is limited and often focused on indirect analysis (i.e., NSSI associated with internet or food addiction or substance use), which supports the mechanism of emotion regulation rather than dependence patterns of NSSI [63,64]. This is different from Fontecilla et al. [65], who share an addictive model of self-injurious behaviors, explaining it with neurobiological and psychological mechanisms. On the one hand, the opioid and dopaminergic systems, as well as the HPA axis, interact in the forebrain and can be activated by psychoactive drugs. On the other hand, the cathartic effect activated by the mobilization of interpersonal support (e.g., medical care, family care) or the emotional outburst of an intolerable psychophysical state contributes to the addiction in self-injurious behaviors.

The comparison between emotional and behavioral components, evaluated by BIS and OSI distinguished in the four subgroups of NSSI patients according to the number of NSSI episodes, the number of injury sites, the number of injury modes and the association with suicide attempts, showed that:

(1) All patients with a tendency to repeat self-injurious behavior (multiple NSSI episodes, multiple sites of injury, multiple modes of injury and with suicide attempts) had item 20 of OSI score significantly higher compared to the other subgroup. Therefore, engaging in NSSI to experience physical pain could be interpreted as an addictive behavior probably related to the increase in dopamine levels after the self-injury, similar to substance use disorders [66]. Once again, the importance of the additive component in non-suicidal self-injurious behavior is evident, especially in its most serious manifestations.

(2) NSSI patients with suicide attempts had significantly higher IER, SS and MI scores than the other subgroup in comparison. Unfortunately, the structure of our study is not able to give any interpretation about the nature and development of these associations. Although a large literature supports the existence of a strong association between emotion regulation and self-injury (NSSI, suicide attempts), this association is established with heterogenic methods, making comparison difficult. Intense negative emotional arousal and difficulty with emotional adjustment appear to be risk factors for suicidal ideation [67]. In contrast, adolescents with a history of NSSI reported significantly lower distress tolerance and greater emotional reactivity than adolescents with suicide attempts [68]. NSSI subjects with and without suicide attempts showed disorganized attachment pathways and impaired reflex functioning [69]. In the end, a model about the role of emotion regulation in NSSI suggested that the relationship between some aspects of emotional dysregulation and suicide attempts may be indirectly modulated by NSSI [70].

Various impulsive traits were consistently associated with self-injury that was classified as non-suicidal related, in part because many self-injury studies have openly excluded suicidal intent. However, there is evidence that subjects who had both NSSI behavior and suicide attempts had significantly higher impulsivity traits than those who had NSSI behavior alone [50,58].

At this stage, we could suppose that NSSI and suicidal behavior are similar but separate phenomena, which can also coexist and relate, activated within a short time period and underpinned by similar mechanisms [16].

## 6. Conclusions

There are some limitations of this study, such as the small sample size, which made comparisons between subgroups difficult, the choice to analyze NSSI in subjects with neuropsychiatric disorders, which certainly influenced the scores obtained in the tests we used and the lack of a control group. Nevertheless, it is clear that there are several factors underlying NSSI, which must necessarily be considered as a multifactorial construct. It is therefore necessary to carefully evaluate all of the involved components (clinical, emotional and behavioral) in the initiation and maintenance of self-injurious behaviors, including addictive components and those associated with the most severe NSSI cases, in order to address appropriate treatment options for each subject (such as dialectical and cognitive–behavioral therapy, mentalization and motivation-based approaches and group therapies) [7,12,15].

Future research should be conducted through longitudinal studies using shared standardized measures and study designs.

## Figures and Tables

**Table 1 brainsci-11-00974-t001:** Socio-demographic and clinical features of the sample.

	*N*	%
Hospitalized patients	3944	100
Patients with NSSI	54	1.36
Males with NSSI	10	18.5
Females with NSSI	44	81.5
Age at the first observation		
12	3	5.6
14	6	11.1
15	13	24
16	9	16.6
17	9	14.8
18	4	7.4
19	2	3.7
20	2	3.7
22	1	1.9
25	2	1.9
28	2	3.7
33	1	1.9
Neuropsychiatric Disorders		
Depressive Disorders	20	37
Trauma and Stressor Related Disorders	12	22.2
Personality Disorders	8	14.8
Feeding and Eating Disorders	4	7.4
Bipolar and Related Disorders	3	5.5
Schizophrenia Spectrum and Other Psychotic Disorders	3	5.5
Anxiety Disorders	2	3.8
Disruptive, Impulse Control and Conduct Disorders	1	1.9
Somatic Symptom and Related Disorder	1	1.9

**Table 2 brainsci-11-00974-t002:** Ottawa Self-injury Inventory Functions scores.

OSI Functions	Minimum Score	Maximum Score	Mean ± SD
Internal Emotion Regulation	0	28	12.52 ± 6.87
Social Influence	0	20	4.60 ± 4.53
External Emotion Regulation	0	12	6.57 ± 3.88
Sensation Seeking	0	14	1.43 ± 3.08
Craving	0	25	13.47 ± 6.35

**Table 3 brainsci-11-00974-t003:** Barratt Impulsiveness Scale 11 scores.

BIS 11	Minimum Score	Maximum Score	Mean ± SD
Cognitive Impulsivity	10	29	19.38 ± 4.40
Motor Impulsivity	12	40	23.13 ± 5.25
Non-Planning Impulsivity	16	37	28.06 ± 5.22
Total Impulsivity	45	90	70.55 ± 11.12

**Table 4 brainsci-11-00974-t004:** Mann–Whitney comparisons between OSI and BIS 11 scores in NSSI patients with an occasional episode (*n* = 36) vs. NSSI patients with multiple episodes (*n* = 18) and rank biserial correlation (rB) as a measure of effect size.

				95% CI for rB	
OSI and BIS 11 Items	W	*p*	rB	Lower	Upper
Internal Emotion Regulation	187.000	0.569	−0.110	−0.447	0.255
Social Influence	165.500	0.438	−0.151	−0.487	0.224
External Emotion Regulation	241.500	0.431	0.150	−0.216	0.479
Sensation Seeking	248.000	0.241	0.181	−0.185	0.503
Craving	128.500	0.001	−0.566	−0.754	−0.291
Cognitive Impulsivity	316.500	0.698	0.069	−0.266	0.389
Motor Impulsivity	276.500	0.712	−0.066	−0.387	0.269
Non-Planning Impulsivity	235.000	0.239	−0.206	−0.501	0.132
Total Impulsivity	262.500	0.522	−0.113	−0.426	0.224

**Table 5 brainsci-11-00974-t005:** Mann–Whitney comparisons between OSI and BIS 11 scores in NSSI patients without suicide attempts (*n* = 27) vs. NSSI patients with suicide attempts (*n* = 27) and rank biserial correlation (rB) as a measure of effect size.

				95% CI for rB	
OSI and BIS 11 Items	W	*p*	rB	Lower	Upper
Internal Emotion Regulation	133.000	0.011	−0.449	−0.680	−0.138
Social Influence	197.500	0.431	−0.141	−0.456	0.205
External Emotion Regulation	170.500	0.095	−0.294	−0.571	0.042
Sensation Seeking	163.000	0.023	−0.325	−0.593	0.008
Craving	187.000	0.006	−0.446	−0.662	−0.161
Cognitive Impulsivity	266.000	0.269	−0.182	−0.465	0.136
Motor Impulsivity	205.500	0.025	−0.368	−0.608	−0.066
Non-Planning Impulsivity	330.000	0.932	0.015	−0.296	0.323
Total Impulsivity	248.000	0.149	−0.237	−0.509	0.078

**Table 6 brainsci-11-00974-t006:** Mann–Whitney comparisons between OSI and BIS 11 scores in NSSI patients with a single injury mode (*n* = 26) vs. NSSI patients with multiple injury modes (*n* = 28) and rank biserial correlation (rB) as a measure of effect size.

				95% CI for rB	
OSI and BIS 11 Items	W	*p*	rB	Lower	Upper
Internal Emotion Regulation	202.500	0.457	−0.135	−0.451	0.212
Social Influence	237.500	0.688	0.075	−0.275	0.407
External Emotion Regulation	213.000	0.622	−0.090	−0.414	0.255
Sensation Seeking	200.500	0.329	−0.143	−0.458	0.204
Craving	227.500	0.029	−0.350	−0.591	−0.051
Cognitive Impulsivity	239.500	0.049	−0.316	−0.566	−0.013
Motor Impulsivity	328.000	0.701	−0.063	−0.360	0.246
Non-Planning Impulsivity	340.500	0.872	−0.027	−0.329	0.280
Total Impulsivity	289.000	0.280	−0.174	−0.455	0.137

**Table 7 brainsci-11-00974-t007:** Mann–Whitney comparisons between OSI and BIS 11 scores in NSSI patients with a single site of injury (*n* = 25) vs. NSSI patients with multiple sites of injury (*n* = 29) and rank biserial correlation (rB) as a measure of effect size.

				95% CI for rB	
OSI and BIS 11 Items	W	*p*	rB	Lower	Upper
Internal Emotion Regulation	210.500	0.528	−0.114	−0.432	0.230
Social Influence	249.500	0.550	0.109	−0.240	0.433
External Emotion Regulation	246.000	0.848	0.036	−0.303	0.366
Sensation Seeking	247.500	0.780	0.042	−0.297	0.372
Craving	260.000	0.177	−0.220	−0.495	0.095
Cognitive Impulsivity	284.500	0.370	−0.147	−0.436	0.169
Motor Impulsivity	344.500	0.846	0.033	−0.278	0.338
Non-Planning Impulsivity	327.500	0.919	−0.018	−0.325	0.292
Total Impulsivity	306.500	0.625	−0.081	−0.380	0.233

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
