# Peer review of "Non-Suicidal Self-Injury: An Observational Study in a Sample of Adolescents and Young Adults"

_brainsci, 2021, doi:10.3390/brainsci11080974_

Round 1
Reviewer 1 Report
Its novelty and contribution to science are limited. I have a few comments that I believe would improve the contribution of this manuscript.
I think the rationale for this study could be strengthened. I did not find the argument about the need for this study (what is new and important compared to existing work) very clear or robust. The authors discuss the reasons for NSSI. I wondered whether their argument could be made clearer with reference to the Nonsuicidal Self-Injury Beliefs Scale (NSIBS), which is I believe the most comprehensive measure of NSSI beliefs in existence. See Siddaway, A. P., Wood, A. M., O’Carroll, R. E., & O’Connor, R. C. (2019). Characterizing self-injurious cognitions: Development and validation of the Suicide Attempt Beliefs Scale (SABS) and the Nonsuicidal Self-Injury Beliefs Scale (NSIBS). Psychological Assessment, 31, 592-608. Some rationale is provided for studying youth but little for researching inpatients.
The psychometric properties of the measures used and a measure of internal consistency need reporting for each measure.
Effect sizes need reporting.
With n = 55, there is little rationale for the 4 subgroups.
Author Response
Please, see the attachment.

Reviewer 2 Report
Main message of the article
The articles present data on Non-Suicidal Self-Injury (NSSI) behavior from a clinical sample of teenagers mainly and young adults. The authors Used the Ottawa Self-Injury Inventory to investigate the affective, behavioral, cognitive, and environmental components of NSSI, and the Barrat Impulsiveness Scale 11 (BIS11) to assess impulsivity. The authors provide insights on NSSI analyzing a sample of young inpatients diagnosed with a mental disorder, and exploring the main emotional and behavioral dimensions.
General Judgment Comments
The style of the manuscript is acceptable, although the text would benefit from a substantial improvement of the language and of the scientific lexicon. The design of the study and the analyses present some fragilities that need to be considered and solved. The title is informative about the main purpose of the study, while in the abstract some information is missing. Tables need to be edit to display a consistent output (i.e., line spacing). In the current version, some major and minor issues preclude the publication of the study at this stage. Please see comments below.
Major Issues
- Please, insert a Table where results from the Mann-Whitney test are shown clearly, so it is easier to follow the results and to detects statistically significant outcomes;
- STATISTICAL ANALYSES: It would be interesting and helpful to report a table with the correlations among the considered subscales and variables. Moreover, the authors should provide information on the internal consistency (i.e., Cronbach’s alpha) for the OSI and the BIS, to prove the reliability of the two questionaries and their subscales;
- Although the main message of the paper is overall intelligible, the English in the present manuscript is not of publication quality and requires major improvement. Authors are required to carefully proofread spell check to eliminate errors (i.e., singular/plural, verb tenses, missing punctuations), refine the lexicon, and reduce the repetition of the same words, which makes the text sound redundant.
To report a few examples:
- Line 35: the authors could use “can be observed” instead of “begins” as NSSI is a phenomenon that might occur, while “begins” has a more deterministic nuance in its meaning.
- Line 38: “there has been debated” is not very clear; authors could either use “it has been debated” or “there has been a debate”.
- Line 42-43: “Individual” and “their” do not match properly;
- Line 45: “or other sharp object”; the authors could either write “another sharp object”, or “other sharp objects”.
These are just a few examples in the first lines. Meticulous proofreading of the whole manuscript would improve the quality of the article, giving greater value to the message of the authors and the findings.
Minor Issues
- In the abstract, the main topic (NSSI) is introduced very briefly, and a few more words might be helpful to better contextualize the issue;
- The present version reports 3 keywords, which are extremely relevant to the topic, but adding a few more keywords (at least 2/3) would also help other researchers and readers to find more easily the paper;
- In some parts, the text would benefit from some further references. For instance:
- Line 56: “An interpretation of NSSI consists in an integrated model…”; is this a reference to a published model (if so, which one?) or it is a model suggested by the authors? Please, disambiguate, preferably using references to existing literature;
- Line 107: add a citation for the validated Italian version of the OSI;
- Line 144: add a reference for the validated Italian version of the BIS (#15 of the reference list);
- Line 221: add a citation to support this sentence;
- Lines 236-239: add a citation to support this sentence;
- Line 273-274: a review recently published discusses the association between social media usage and psychiatric disorders in adolescents, including NSSI. Following the reference, in case of the authors would consider adding it to the manuscript to strengthen their example:
Cataldo, I., Lepri, B., Neoh, M. J. Y., & Esposito, G. (2021). Social media usage and development of psychiatric disorders in childhood and adolescence: A review. Frontiers in Psychiatry, 11, 1332.
- Line 108: the authors state that they will be using only some items from the OSI; why were the other items excluded? Please, clarify this decision;
- Lines 227-231: the explanation of potential brain alteration related to NSSI is very brief and a bit misleading: authors should either expand this explanation or delete it;
- Please check the line spacing in the tables to make the format consistent throughout the paper;
- Please add a section after “Authors Contribution” reporting the list of abbreviations. Since there are many abbreviations, it would be useful to have a section where the reader can easily find the full meaning of the acronyms, instead of search throughout the paper. Some abbreviations, such as “CIm, MIm, NPIm”, are not explained and their meaning might be misunderstood, compromising the comprehension of results.
Final comments
Authors are required to fix the issues reported following the suggestions provided and resubmit the paper for further consideration before publication.
Author Response
Please, see the attachment

Round 2
Reviewer 1 Report
The tables were difficult to read and seem to be missing degrees of freedom column.
The standard of English could be improved
No study ever 'proves' anything. This study 'suggests' 'found' 'indicated
I still think the rationale for this study - its novelty and importance - could be strenghtened
Author Response
The tables were difficult to read and seem to be missing degrees of freedom column.
The standard of English could be improved.
No study ever "proves" anything. This study "suggests" "found" indicated.
I still think the rationale for this study - its novelty and importance - coul be strenghtened.
Dear Reviewer,
we greatly appreciated your comments. We hope that you will consider our changes satisfactory to increase the quality of our manuscript.
As you suggested, we improved the standard of English and comprehensibility of the new tables for the readers; "df" was a typo.
In the Discussion section we removed and replaced the term "proves".
We modified the Introduction section to improve and clarify the rationale for this study.
Kind regards,
Emilia Matera
Reviewer 2 Report
Good study
Author Response
Dear reviewer,
thank you for your precious and kind support.
Yours sincerely,
Emilia Matera